# Identification of Virulence Genes Associated with Pathogenicity of Translocating *Escherichia coli* with Special Reference to the Type 6 Secretion System

**DOI:** 10.3390/microorganisms12091851

**Published:** 2024-09-06

**Authors:** Behnoush Asgari, Jarred R. Burke, Bonnie L. Quigley, Georgia Bradford, Eva Hatje, Anna Kuballa, Mohammad Katouli

**Affiliations:** 1Centre for Bioinnovation, University of the Sunshine Coast, Maroochydore DC, QLD 4558, Australia; basgari@usc.edu.au (B.A.); jarred.burke@servatus.com.au (J.R.B.); bonnie.quigley@usc.edu.au (B.L.Q.); georgiabradford98@gmail.com (G.B.); akuballa@usc.edu.au (A.K.); 2Servatus Biopharmaceuticals, Coolum Beach, QLD 4573, Australia; 3Thompson Institute, University of the Sunshine Coast, Maroochydore DC, QLD 4558, Australia; 4Centre for Immunology and Infection Control, Faculty of Health, School of Biomedical Sciences, Queensland University of Technology (QUT), Brisbane, QLD 4000, Australia; e.hatje@qut.edu.au; 5School of Health, University of the Sunshine Coast, Maroochydore DC, QLD 4558, Australia

**Keywords:** translocating *E. coli*, T6SS, virulence genes, Caco-2:HT29-MTX, gene expression

## Abstract

Recent genomic characterisation of translocating *Escherichia coli* HMLN-1 isolated from mesenteric lymph nodes (MLNs) and blood of a patient with a fatal case of pancreatitis revealed the presence of a type 6 secretion system (T6SS) that was not present in non-translocating *E. coli* strains. This strain was also genomically similar to adherent-invasive *E. coli* (AIEC) LF82 pathotype. We aimed to identify the role of T6SS-1 in the pathogenesis of this strain and other pathogenic *E. coli*. The HMLN-1 strain was initially tested for the presence of six virulence genes (VGs) associated with AIEC strains and an iron sequestering system. Additionally, HMLN-1’s interaction with a co-culture of Caco-2:HT29-MTX cells and its intra-macrophagic survival was evaluated. We subsequently screened a collection of 319 pathogenic *E. coli* strains isolated from patients with urinary tract infection (UTI), diarrhoea, inflammatory bowel disease (IBD) and septicaemia for the presence of T6SS-1 and its expression related to adhesion, invasion and translocation via the above co-culture of the intestinal cell lines. The results showed that HMLN-1 harboured four of the AIEC-associated VGs (*dsb*A, *htr*A, *omp*C and *afa*C). Screening of the pathogenic *E. coli* collection detected the presence of the T6SS-1 genes in septicaemic and UTI *E. coli* strains at a significantly higher level than diarrhoea and IBD strains (*p* < 0.0001). The high expression of T6SS-1 in *E. coli* HMLN-1 upon adhesion and invasion, as well as its high prevalence among extra-intestinal *E. coli* strains, suggests a role for T6SS-1 in the pathogenesis of translocating *E. coli*.

## 1. Introduction

Bacterial translocation is the migration of viable bacteria and their products across the gut epithelium to the bloodstream to cause systemic infection [1,2]. Under normal physiological conditions, the gastrointestinal (GI) tract acts as an immunological barrier preventing translocation of enteric bacteria into the bloodstream [3]. However, when the normal ecological balance of the gut is disturbed, e.g., during antibiotic treatment [4,5] or host stress [6,7], certain bacteria can increase in number, adhering to the gut epithelium and crossing the gut barrier to appear in otherwise sterile extra-intestinal sites [8].

For a bacterium to translocate across the gut epithelium, it must first establish adhesion with the host cell via extracellular fimbrial or afimbrial adhesins [9,10]. These virulence factors are commonly associated with extra-intestinal pathogenic *E. coli* (ExPEC) strains such as uropathogenic *E. coli* (UPEC) and sepsis-associated *E. coli* [10,11]. Following intimate cell–cell adhesion through interaction with membrane-bound receptors on the host cell, ExPEC strains can invade cells through the release of toxins or invasins [10] to promote cytoskeleton rearrangement and internalisation [12]. However, some strains of *E. coli* have been shown to translocate efficiently through the gut epithelium by invading intestinal cells via a different ‘triggering’ pathway using membrane-bound vesicles for transport [13,14]. These strains have been referred to as translocating *E. coli* (TEC) [15]. Following cellular invasion by the pathogen and transcytosis to the lamina propria, the bacterium must then survive the host immune response, including uptake by phagocytic cells [13], and replicate in macrophages [16].

We have previously reported isolation of several translocating *E. coli* strains from humans and animals under different physiological stressors [17,18,19]. The human *E. coli* strain, HMLN-1, was isolated from a patient with fatal pancreatitis and has been found in previous studies to efficiently translocate in both animals and cell culture models representing the gut epithelium [9,13,20,21]. Ramos et al. (2011) tested the presence of 44 virulence genes (VGs) commonly reported among pathogenic *E. coli* and found only a *kps*MT group III polysialic capsule synthesis gene in *E. coli* HMLN-1, leaving the pathogenic mechanism of HMLN-1 largely unknown [13]. Through complete genomic sequencing of *E. coli* HMLN-1, we identified the presence of a GI-*arg*U pathogenicity island (PAI), a type 6 secretion system (T6SS) and a genomic backbone similar to the known adherent-invasive *E. coli* (AIEC) strain LF82 [15]. AIEC strains are commonly found to be associated with the gut mucosal epithelium and carry several genotypic and phenotypic characteristics that are not usually seen in other *E. coli* pathotypes [22].

In this study, we initially tested the presence of virulence genes associated with AIEC in HMLN-1, as well as the ability of HMLN-1 to interact with the gut epithelium. Our previous studies used monoculture cell lines such as Caco-2 or HT-29 cells to represent the gut epithelium [8,13,21,23]. Recent advances in cell culture assays have suggested that monoculture cell lines may not fully represent the gut intestinal epithelium. For instance, while Caco-2 cells are polarised cells with tight junctions and microvilli, they do not produce mucus, which is an important characteristic of human intestinal mature enterocytes. To overcome this shortcoming, another cell line, HT-29, has been treated with methotrexate (MTX) to facilitate mucin secretion [24,25,26]. Mucus-secreting HT29-MTX subclones have previously been isolated and characterised regarding tight junction formation, development of confluent monolayers and production of mucin [27]. Therefore, we investigated the interaction of HMLN-1 with a co-culture of Caco-2 and HT29-MTX cells that reflects a more similar cellular component of the intestine than each of the monoculture cell lines alone. In addition, because of our recent finding that the *E. coli* HMLN-1 strain also carries T6SS-1 [15], which is not commonly found among other human pathotypes of *E. coli*, we hypothesised that T6SS-1 plays a role in the pathogenesis of extra-intestinal *E. coli* (including their translocation across the gut epithelium). Therefore, the second goal of this study was to investigate the presence of T6SS-1 in a collection of *E. coli* isolated from intestinal and extra-intestinal infections and assess the expression of T6SS-1 genes within HMLN-1 upon interaction with this co-culture of Caco-2 and HT29-MTX cells as a model of the gut epithelium.

## 2. Materials and Methods

### 2.1. E. coli Strains

The *E. coli* HMLN-1 strain was originally isolated from the blood and mesenteric lymph nodes (MLNs) of a patient with a fatal case of pancreatitis [28] and was found in several studies to be a professional translocator [13,21,29,30]. The strain was stored in tryptone soy broth (TSB) with 20% glycerol at −80 °C. Strain purity was tested and maintained through streaking on MacConkey agar no. 3 (Oxoid, Thermo Fisher Scientific, Waltham, MA, USA) and incubation at 37 °C for 24 h. Isolated colonies were selected and grown on nutrient agar (NA) as working cultures for use in cell culture assays and for DNA and/or RNA extraction [29,31]. A collection of 319 *E. coli* strains isolated from patients with intestinal and extra-intestinal infections were also included in this study and tested for the presence of T6SS-1. These included strains isolated from adults (*n* = 40) and children (*n* = 75) with hospital-acquired (HA) urinary tract infection (UTI), children with diarrhoea (*n* = 30), adults with community acquired (CA)-UTI (*n* = 75), adults with IBD (*n* = 25) and finally adults with hospitalised septicaemia without UTI (*n* = 74). The clonality of strains was determined using a combination of PhP biochemical fingerprinting and RAPD-PCR typing methods in previous studies [32,33]. Strains having the same PhP/RAPD-PCR patterns were regarded as identical and assigned to the same common types (CTs), while strains that did not share the same PhP/RAPD-PCR types were named as single types (STs). In all, 37 CTs and 95 STs were identified within the collection. Representative strains of each CT were selected and, together with all 95 STs, tested for the presence of T6SS using *clp*V and *vgr*G genes (Appendix A).

### 2.2. Testing for Adhesion and Invasion

A co-culture of human colonic adenocarcinoma cell lines Caco-2 (ATCC = HTB 37) and HT29-MTX (ATCC = HTB 38) (in a ratio of 9:1, respectively) was used as a model of the human gut epithelium to investigate the ability of *E. coli* HMLN-1 to diffusely adhere (DA) to, invade and translocate via intestinal epithelial cells. Both cell lines were initially grown separately to >75% confluence in Eagle’s Minimum Essential Medium (EMEM; Gibco, Grand Island, NY, USA), supplemented with 20% (*v*/*v*) foetal bovine serum (FBS; Gibco) for Caco-2 cells and 15% for HT29-MTX cells in the presence of 1% (*v*/*v*) penicillin/streptomycin (Thermofisher, Scoresby, VIC, Australia). The growth condition was 37 °C and 5% CO_2_, with changes in culture media occurring at 48 h intervals. For the adhesion assay, cells were seeded into an 8-well chamber slide (Lab-Tek) at a density of 4 × 10^4^ cells/cm^2^ in a cell suspension ratio of 9:1 Caco-2:HT-29 MTX and grown to >75% confluence [34]. The *E. coli* strain HMLN-1 and the negative control strain *E. coli* JM109 (a laboratory K12 strain) were cultured overnight in Luria–Bertani (LB) broth at 37 °C with an agitation of 130 rpm. Bacterial suspensions were centrifuged at 4000 rpm for 12 min, the pellet resuspended in sterile phosphate-buffered saline (PBS) and the concentration of bacteria adjusted to 1 × 10^9^ CFU (OD_600nm_ = 1.0). These suspensions were further diluted to 10^8^ CFU/mL, and 100 µL of the suspension was inoculated into appropriate cell culture wells containing 300 µL of growth medium without penicillin/streptomycin. The chamber slides were then incubated at 37 °C for 120 min, before washing with PBS three times to remove non-adhering bacteria. Adhered bacteria were fixed to the slide with 95% (*v*/*v*) ethanol for five minutes, stained for 15 min with 10% (*v*/*v*) Giemsa stain and observed under a light microscope. The percentage of cells showing bacterial adherence (i.e., percentage of colonisation of the HMLN-1 strain) was determined by observing bacterial presence/absence on 100 randomly selected cells, and the number of bacteria adhering per cell was calculated by counting 25 randomly selected cells showing adhesion. The results were expressed as mean ± standard error of the mean (SEM).

For the invasion assay, the co-culture of Caco-2:HT-29 MTX cells was seeded at a density of 2 × 10^4^ cells/cm^2^, and the cells were grown to confluence in a flat-bottomed 96-well plate. To determine the multiplicity of infection (MOI), three wells with confluence growth were washed, detached using 0.25% Trypsin-EDTA (Gibco, USA) for 5 min and the cells were counted using an improved Neubauer hemocytometer, C-Chip (ProSciTech, Kirwan, QLD, Australia), to determine average cell numbers per well. *E. coli* strains were also prepared as described for adhesion assay, and their concentration was adjusted to give an MOI of 100 (100 bacteria per cell) after inoculating 100 µL of bacterial suspension into corresponding wells containing 100 µL of growth medium without penicillin/streptomycin. After 2 h of incubation at 37 °C, the cells were washed with PBS three times to remove non-invading bacteria. The wells were then treated with 200 µL of EMEM/FBS supplemented with 100 µg/mL gentamicin (Thermofisher, Australia) before incubation for another hour to kill extracellular bacteria. The cells were then washed three times with PBS and lysed with 200 µL of 0.1% (*v*/*v*) Triton-X-100 (Sigma-Aldrich, St. Louis, MO, USA) to release invading bacteria. Following 15 min of incubation at 37 °C, the lysate was serially diluted 10-fold, and 100 µL was plated onto MacConkey agar no. 3 (Oxoid). The agar plates were incubated at 37 °C for 24 h, and the number of invading bacteria was calculated and expressed as mean ± SEM following correction for the dilution factor. The *E. coli* strain JM109 was used as a negative control.

### 2.3. Translocation Assay

For the translocation assay, the co-culture of cells was seeded with 4 × 10^4^ cells into a 0.8 µM pore size Millicell (Millipore, Burlington, MA, USA) insert containing 300 µL of EMEM and 20% FBS with antibiotics. Each insert was placed within a well of a 24-well plate containing 600 µL of EMEM without antibiotics. Cells were grown to confluence over 21 days to allow microvilli to form on the monolayer surface. The integrity of the cell lines grown in Millicell inserts was measured using transepithelial electrical resistance (TEER) between the inner and the outer chambers (TEER value of 590–662 Ωcm^2^) as described before [21]. Media within the inner wells were replaced with EMEM without antibiotics before addition of 100 µL of bacterial suspension, prepared as described above, and the microplate was incubated at 37 °C. Samples (100 µL) were collected from the outer well after 15, 30, 60 and 120 min of incubation, serially diluted and spread on MacConkey agar. After each sampling, 100 µL of fresh media was added to the outer well to maintain the volume at 600 µL. The MacConkey agar plates were incubated for 24 h at 37 °C, after which the number of translocating bacteria was calculated. Negative control strains included in translocation studies were *E. coli* JM109, *E. coli* KIC-1 (which has been shown to be capable of translocation in laboratory animals [21,23] but lacking T6SS-1 [15]), *E. coli* 73–89 (capable of adhering to the gut epithelium in animal models but not translocating [8]) and *E. coli* 46-4 (a non-adhering/non-translocating strain in animal models [15,21]).

### 2.4. DNA Extraction

Chromosomal DNA of *E. coli* strains was extracted using the boiling method, as previously described [35]. Briefly, isolated bacterial colonies were suspended in 150 µL of 0.6× Tris-EDTA (TE) buffer, before boiling in a dry-block heater for 15 min at 100 °C. The sample were then centrifuged for 10 min at 13,500 rpm. The supernatant was removed and stored at −20 °C for use in PCR detection of VGs.

### 2.5. PCR Detection of VGs

Due to the genetic similarity of HMLN-1 to AIEC strains with the ability to adhere to and invade through the gut epithelium, we investigated the presence of six VGs associated with the AIEC pathotype in HMLN-1 using the method described previously [36]. These included genes encoding production of long polar fimbriae (*lpf*A); outer membrane protein C (*omp*C); afimbrial adhesin outer membrane protein (*afa*C); intra-macrophagic survival and replication genes, including high temperature requirement A (*htr*A) and oxidoreductase disulphide bond A (*dsb*A); and a gene encoding colibactin (*clb*A), which is a genotoxin located on the pks-PAI [36,37,38]. The primer sequences used for these genes and their base pair fragment sizes are presented in Appendix A. Previously isolated laboratory *E. coli* strains KIC-2, RBH128, RBH123 and AAA-172 were used as positive controls for *lpf*A, *htr*A, *dsb*A, *afa*C, *omp*C and *clb*A genes. PCR reactions were performed using 1 × PCR reaction buffer (Bioline, London, UK), 0.4 mM dNTPs, 3 mM MgCl2, 0.3 µM of forward and reverse primer (Invitrogen), 0.2 U Bioline Taq DNA polymerase, 2 µL of template DNA and sterile water to a final reaction volume of 25 µL. Sterile MilliQ water was used as a no-template control.

All PCR amplifications were conducted in a T100 Thermal Cycler (Bio-Rad, Hercules, CA, USA) using cycling conditions described in Appendix A. Amplified PCR products were observed after separation in 1.5% agarose gel with 0.6x TBE buffer for 60 min at 100 volts and stained with ethidium bromide. PCR products were visualised using the Syngene GeneGenius Gel Light Imaging System. The HMLN-1 strain was also tested for the presence of five iron acquisition genes associated with pathogenic *E. coli*. These included genes encoding for aerobactin siderophore receptors (*iut*A) [39], a catechole siderophore receptor (*iroN_E. coli_*) [40], yersiniabactin receptor (*fyu*A), haemolysin-A (*hly*A) [39] and iron-regulated gene (*ire*A) [41] (Appendix A). Laboratory *E. coli* strains RBH2, RBH4 and RBH8 were used as positive controls for amplification of *hly*A, *iut*A, *fyu*A, *ire*A and *iroN_E. coli_*, with MilliQ water acting as a negative control. PCR reactions were performed using 1× PCR reaction buffer (Bioline); 0.4 mM dNTPs; 3 mM MgCl_2_; 0.72 µM of forward and reverse primer for *hly*A, *iut*A and *fyu*A (Invitrogen); 1 µM of forward and reverse primer for *ire*A and *iroN_E. coli_*; 0.2 U Bioline Taq DNA polymerase; 2 µL of template DNA; and sterile water to a final reaction volume of 25 µL. Cycling conditions for PCR amplification are described in Appendix A. Amplified products were separated as described above.

### 2.6. PCR Detection of Type 6 Secretion System

All *E. coli* strains were also tested for the presence of two genes associated with T6SS-1 in *E. coli*. These included genes encoding for a secreted protein of T6SS-1 acting as an effector, valine-glycine repeat-G (*vgr*G) [42] and a T6SS-1 ATPase (*clp*V) [43]. Primer sequences used for these assays are given in Appendix A. The laboratory *E. coli* strain KIC-2, previously found to be positive for T6SS-1 [15], was used as a positive control, while the *E. coli* strain KIC-1 showing lack of T6SS-1 [15] was used as a negative control. PCR reactions were performed using 1× Qiagen Hotstart MasterMix, 0.5 µM forward and reverse primers and 2 µL of template DNA, made up to a final volume of 25 µL with sterile MilliQ water. Cycling conditions for PCR amplification have been presented in Appendix A. Amplified products were observed as described above. PCR amplification of the *vgr*G and *clp*V T6SS-1 genes were confirmed with Sanger Sequencing by Macrogen (Seoul, Republic of Korea) using Staden Package software (https://sourceforge.net/projects/staden/, accessed on 2 September 2024). NCBI BLAST analysis of the resultant sequences confirmed an 89% match with the sequences encoding the T6SS-1 tip protein *vgr*G in *E. coli* (accession number: WP_122997667.1) and a 99.3% match for T6SS-1 ATPase *tss*H (*clp*V) in *E. coli* (accession number: WP_021522427.1).

### 2.7. Gene Expression

The HMLN-1 strain was initially grown overnight in Luria–Bertani (LB) broth at 37 °C with an agitation of 130 rpm. Bacterial suspensions were then centrifuged at 4000 rpm for 12 min, the pellet resuspended in phosphate-buffered saline (PBS) and the concentration adjusted to 1 × 10^9^ CFU (OD600 nm = 1.0). Half of this suspension was used for isolating bacterial RNA before HMLN-1 was added to the cell co-culture (control group). The other half was used for measuring the expression of T6SS-1 after interaction with the cells. The cell co-culture was grown in a 24-well microplate to confluence and was inoculated with a suspension of the HMLN-1 strain as described for the invasion assay. Before inoculating the bacterium, the cell media were replaced with antibiotic-free medium, and the microplate was incubated at 37 °C for 120 min. Following the incubation, the bacterial suspensions within all wells were removed, pooled, centrifuged and used for RNA extraction (adhesion). The cells within each well were then washed with PBS, and the cells were lysed with 200 µL of 0.1% (*v*/*v*) Triton-X-100 (Sigma-Aldrich) to release invading bacteria, which were subsequently used for RNA extraction (invasion).

### 2.8. RNA Extraction

All three bacterial samples were subjected to RNA extraction using the RNeasy Mini Kit (Qiagen, Clayton, VIC, Australia) with a pre-lysozyme extraction of the cell wall of bacteria according to the manufacturer’s instructions. The 700 µL lysates were transferred to an RNeasy Mini spin column placed in a 2 mL collection tube and centrifuged for 15 s at ≥10,000 rpm. The flow-through was discarded, and the RNA was eluted in RNAse-free water. The quantity and purity were checked using a nanodrop and gel electrophoresis.

The extracted RNAs in both adhesion and invasion treatments were reverse transcribed using the SensiFast cDNA Synthesis kit (Bioline) and RNase Inhibitor (RNeasy Mini Kit) (Qiagen, Australia) in a final volume of 20 μL according to the manufacturer’s instructions. The synthesised cDNA (200 ng) was subjected to qPCR amplification incorporating 1 × of QuantiTect SYBR Green PCR Master Mix (Qiagen) into the reaction tube and 300 nM of each forward and reverse primer (Table 1). The primers were designed using NCBI Primer Blast. qRT-PCR was carried out using the QuantiTect SYBR green PCR kit (Qiagen) following the manufacturer’s protocol. qRT-PCR reactions were run in a Rotor-gene^®^ Q 76 machine (Qiagen) with the following cycling conditions: polymerase activation at 95 °C (15 min), 35 cycles of denaturation at 94 °C (15 s), annealing at 55 °C (30 s) and extension at 72 °C (90 s) followed by melt curve analysis. The fold change in RNA levels was calculated using the ΔΔCt method. The relative expression of each gene was normalised using the two reference housekeeping genes, glyceraldehyde 3-phosphate dehydrogenase (*gap*A) and aerobic respiration control A (*arc*A) [44]. Expression profiles of the T6SS-1 gene and the two housekeeping genes were determined in three biological replicates, each tested in triplicate.

### 2.9. Statistical Analysis

Statistical analysis was performed using GraphPad Prism version 8 for Windows (GraphPad Software, San Diego, CA, USA). Fisher’s exact test was used to compare the prevalence of T6SS-1 VGs among the tested groups and the percentage of cell colonisation in adhesion assays. Unpaired t-tests were used to compare differences in adhesion and invasion data. Differences between the levels of translocation among the strains were determined using an ordinary one-way analysis of variance (ANOVA), with a Dunnett’s multiple comparisons test for comparison to HMLN-1. Differences were considered statistically significant if *p* < 0.05.

## 3. Results

### 3.1. Adhesion and Invasion Assays

The percentage of colonisation of the co-culture cells by the HMLN-1 strains differed significantly (*p* < 0.0001) from the control strain JM109 (Table 1). Compared to the non-TEC control strain JM109 (which showed very little adherence to the co-culture cells, 2.1 ± 0.2 CFU/cell), the HMLN-1 strain showed significantly more diffuse adherence to the co-culture cells (5.7 ± 0.4 CFU/cell, *p* < 0.0001). While both HMLN-1 and JM109 strains demonstrated the ability to invade the co-culture cells, the HMLN-1 strain invaded significantly more than the JM109 strain (*p* = 0.0012) (Table 1).

### 3.2. Translocation Assays

The cumulative number of translocating HMLN-1 bacteria over the two-hour translocation assay was significantly higher than the control strains KIC-1 (known to translocate but not carry the T6SS system gene), 73–89 (adhering but non-translocating *E. coli* strain), 46-4 (non-adhering, non-translocating *E. coli* strain) and JM109 (negative control strain JM109) (*p* = 0.0243) (Figure 1).

### 3.3. Identification of VGs

The HMLN-1 strain was found to harbour four of the six AIEC-associated VGs investigated (*htr*A, *omp*C, *dsb*A and *afa*C; Table 2). Similar results were obtained with *E. coli* KIC-1, which is known to translocate but not carry the T6SS-1 gene. PCR amplification of iron sequestration genes showed HMLN-1 possessed four iron sequestration genes (*hly*A, *fyu*A, *ire*A and *iroN_E. coli_*) compared with the KIC-1 and 73–89 strains, which possessed only two (Table 2). The T6SS-1 genes *vgr*G and *clp*V were only detected in HMLN-1.

The role of T6SS-1 in adhesion and invasion of HMLN-1 was tested by measuring the differential expression of the genes *vgr*G (Figure 2a) and *clp*V (Figure 2b) before and after HMLN-1 interaction with the cell co-culture. The results showed a significant increase in the expression of both genes after adhesion and invasion of HMLN-1 to the cell co-culture (Figure 2).

### 3.4. Prevalence of T6SS-1 among Clinical Strains of E. coli

Initial typing of the 319 *E. coli* strains isolated from patients with intestinal (IPEC) and extra-intestinal *E. coli* strains (ExPEC) grouped them into 37 C-types and 95 S-types (Appendix A). A representative of each C-type was randomly selected, and all 95 S-types (in total, 132 strains) were tested for the presence of the *clp*V and *vgr*G genes. Within the IPEC groups of *E. coli* strains, 3 out of 30 strains (10%) from children with diarrhoea and 8 out of 25 strains from the IBD group (32%) had detectable T6SS-1 genes. In contrast, the ExPEC group carried a significantly (*p* < 0.0001) higher number of strains with T6SS-1 genes, although the percentages of strains carrying T6SS-1 genes varied among the ExPEC groups (Table 3). There was also no significant difference in the prevalence of detectable T6SS-1 genes between all UTI strains (CA-UTI and HA-UTI; *n* = 190) and septicaemic *E. coli* strains (Table 3).

## 4. Discussion

There is very little information concerning the virulence factors and mechanisms used by *E. coli* strains to translocate through the gut epithelium and survive the host immune response to cause septicaemia. Further, lack of treatment and misdiagnosis of TEC has led to a substantial and increasing mortality rate amongst patients with septicaemia [15,45,46]. Some ExPEC strains, such as neonatal meningitis *E. coli* (NMEC) and uropathogenic *E. coli* (UPEC), have been shown to adhere to and invade epithelial and endothelial cells to cause infection [47,48]. However, the mechanism of pathogenicity of TEC has not been fully elucidated. We previously sequenced the genome of four TEC strains, including HMLN-1, and showed that three of these strains possess a T6SS-1 system which is not commonly found in other human *E. coli* pathotypes [15]. Our previous studies, as well as others, have also shown that to translocate, *E. coli* strains must increase in number and adhere to the gut mucosal epithelium [8,49,50]. Since AIEC strains are found to be closely associated with the gut mucosal membrane, we postulated that TEC strains may have similar virulence characteristics to AIEC. Therefore, we initially tested the HMLN-1 strain for the presence of AIEC- and T6SS-1-associated genes. Consistent with our hypothesis, *E. coli* HMLN-1 was found to harbour several AIEC-associated VGs, including *omp*C, *afa*C, *htr*A and *dsb*A. Previous studies have found that co-expression of the VGs *htr*A and *dsb*A is responsible for intra-macrophagic survival in AIEC strains and promoting replication within the acidic phagolysosome environment of phagocytic cells [51]. Here, we showed that *E. coli* HMLN-1 also possessed these two genes to survive phagocytosis and replicate within macrophages. Interestingly, HMLN-1 did not appear to possess the *lpf*A gene which encodes long polar fimbriae, a mechanism used by AIEC to translocate across M cells and interact with Peyer’s patches [52]. This suggests that HMLN-1 harbours another mechanism for translocation across the gut epithelium.

Using HT-29 [47] or Caco-2 [21] cell lines individually, we previously showed that the HMLN-1 strain can adhere and translocate in these in vitro models. In this study, we employed a co-culture of these two cell lines to assess the interaction of HMLN-1 and other control strains with a more representational gut epithelial model. HT29-MTX is an adaptation of HT-29 with methotrexate (MTX) to produce mucin [53]. This co-culture model reflects the cellular components of the intestine more effectively than monocultures and has been used in several studies [54,55]. In the present study, we found that the translocation pattern of HMLN-1 was consistent with results previously obtained from using monocultures of Caco-2 or HT-29 cells on this strain. Interestingly, the TEC strain KIC-1, which was consistently able to translocate in our previous monoculture experiments [21,47], failed to translocate in our co-culture of cells. Genomic analysis of KIC-1 revealed that it did not possess T6SS-1 [15], suggesting that in a more realistic model of the human gut epithelium, T6SS may play an important role in translocation of *E. coli* in humans.

We were also interested to see the prevalence of T6SS among other pathogenic *E. coli* isolated from patients with intestinal or extraintestinal infections. We found that T6SS genes were detectable in all tested bacterial groups, with the highest prevalence observed among UPEC and septicaemic strains.

To identify the genetic variations that might be related to adhesion and translocation across epithelial cells, Bachmann et al. (2015) compared the genomes of HMLN-1 and KIC-1 strains to those of two non-TEC strains and 41 publicly accessible *E. coli* genomes (representing all the main pathotypes). Interestingly, the genomes of the AIEC strains LF82, NRG857C and UM146 shared the most similarities with the genomic backbone of *E. coli* HMLN-1 genomes [15].

This study found the HMLN-1 strain also had detectable VGs encoding outer membrane protein C and afimbrial adhesins. Both *omp*C [48] and *afa*C have been shown to be associated with adhesion and invasion of AIEC and DAEC strains [36] and may have a similar role in adhesion of the HMLN-1 strain. The lack of a detectable *clb*A gene (encoding for production of colibactin [56]) in the HMLN-1 strain was associated with a lack of any observed cytopathic effects on co-culture cells or disruption of the cell monolayer and tight junctions (as detected by a stable trans-epithelial electrical resistance reading throughout translocation assay [9,21,47]).

The ability for bacteria to sequester enough iron for survival is linked with virulence. To facilitate iron acquisition from the iron-limited environment of the host, *E. coli* strains may possess different iron acquisition mechanisms [57]. HMLN-1 carried four of five tested iron sequestration genes including *hly*A and *fyu*A, which encode for α-haemolysin and siderophore yersiniabactin, respectively. These genes have been identified in septicaemic *E. coli* strains as virulence mechanisms, allowing for the breakdown of erythrocytes to release heme groups [58] and the uptake of iron into the bacteria [59]. The *iroN_E. coli_* and *ire*A genes that were also identified in *E. coli* HMLN-1 have been associated with ExPEC survival in iron-deficient environments; however, they are not necessary for bacterial translocation [39,40]. These findings collectively suggest that translocating strains such as HMLN-1 can diversify their iron sequestration abilities to secure iron from the iron-deficient environment of the bloodstream after they translocate.

To identify the importance of the T6SS-1 effector genes *clp*V and *vgr*G in translocation, we tested the expression of these genes upon interaction with the co-culture cells. A high level of expression of these genes was observed in HMLN-1 after both adhesion and invasion with the co-culture cells compared to when the HMLN-1 strain was grown only in TSB medium. These data suggest that T6SS-1 may be involved in translocation of the HMLN-1 strain across the co-culture cell layer by the action of the T6SS-1 effector genes *clp*V and *vgr*G. These genes were found within the first loci of T6SS-1 by whole-genome sequencing of TEC strains [15]. Interestingly, another TEC strain, KIC-1, which was isolated from the rats subjected to non-lethal haemorrhage and has been shown to translocate in monocultures of either HT-29 or Caco-2 cells [21,48], did not carry T6SS-1 [15] and did not translocate in our co-culture cell model. T6SS has been regarded as a virulence factor among bacteria and designed to deliver effector proteins into target cells in a one-step procedure [60,61,62]. Its contracted sheath is disassembled and recycled by a dedicated ATPase identified as ClpV, which provides energy for the operation of T6SS [60,63]. The 18 genes located within the GI-*arg*U region in the HMLN-1 strain encode all the structural proteins of T6SS and the two T6SS-associated secreted proteins, haemolysin-coregulated protein (Hcp) and valine-glycine repeat G (VgrG) [15]. Within the scope of the current research, we were not yet in a position to perform gene deletion experiments to validate our findings of the importance of the T6SS system in *E. coli* translocation of these strains. The next stage of the project will use gene deletion methods to knock out *vgr*G and *clp*V in *E. coli* HMLN-1 and determine the effect on its ability to translocate. Our search for the presence of T6SS-1 among the collection of *E. coli* isolated from the intestinal and extra-intestinal sources showed a significantly higher number of T6SS-1 among *E. coli* strains isolated from UTI and septicaemia groups compared to intestinal pathogenic strains. We previously showed that the intestinal tract of healthy individuals is an important reservoir for carriage of UPEC [29]. Similar findings have also been reported by others [64,65]. However, we have also showed that 94% of *E. coli* strains isolated from the blood (i.e., septicaemic strains) and urine (i.e., UPEC strains) of the same patients were identical and capable of translocating via HT-29 cells that represent the gut epithelium [9].

In view of the high prevalence of T6SS-1 genes among UPEC strains capable of translocating via the gut epithelium, we propose that ExPEC strains, while carrying their specific virulence properties that allow them to colonise the urinary tract and cause UTI, can also adhere to and translocate via the gut epithelium to cause septicaemia if they also carry T6SS-1. This is further supported by our previous findings that dominant *E. coli* strains isolated from the faeces of healthy individuals that contain virulence genes of UPEC can also adhere to and invade epithelial cells and translocate across Caco-2 cells to cause bloodstream infection [20,30,47].

## 5. Conclusions

In conclusion, the increased expression of *clp*V and *vgr*G genes during adhesion and invasion of the HMLN-1 strain suggests a role in translocation of this strain across the gut epithelium. This study also highlights the importance of the use of Caco-2:HT29-MTX co-culture as a better model of the gut epithelium for translocation studies as the KIC-1 strain, which was void of T6SS-1, did not translocate in this model. The presence of other virulence genes involved in adhesion (i.e., *afa*C) and intra-macrophagic survival (i.e., *htr*A) in HMLN-1 may provide additional roles in translocation of this strain. The high prevalence of T6SS-1 among the ExPEC strains that carry genotypic characteristics of UPEC may also suggest that these strains are capable of translocation across the gut epithelium. This, however, must be verified by testing the ability of ExPEC strains to translocate via the gut epithelium using the co-culture of Caco-2:HT29-MTX.

## Figures and Tables

**Figure 1 microorganisms-12-01851-f001:**
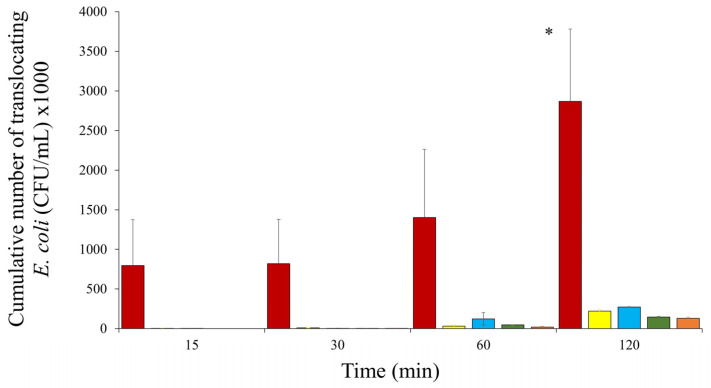
Cumulative number of translocating *E. coli* (mean ± SEM) of HMLN-1 and control strains over 120 min through a co-culture of Caco-2 and HT29-MTX cell lines. Strains tested were HMLN-1 
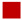
, JM109 
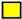
, 46-4 
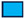
, 73–89 
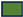
 and KIC-1 
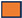
. Only significant values are shown. * At two hours, *E. coli* HMLN-1 vs. KIC-1, JM109, 46-4 and 73–89 (*p* = 0.0243). All tests were conducted in triplicate, and the results were expresses as mean ± SEM.

**Figure 2 microorganisms-12-01851-f002:**
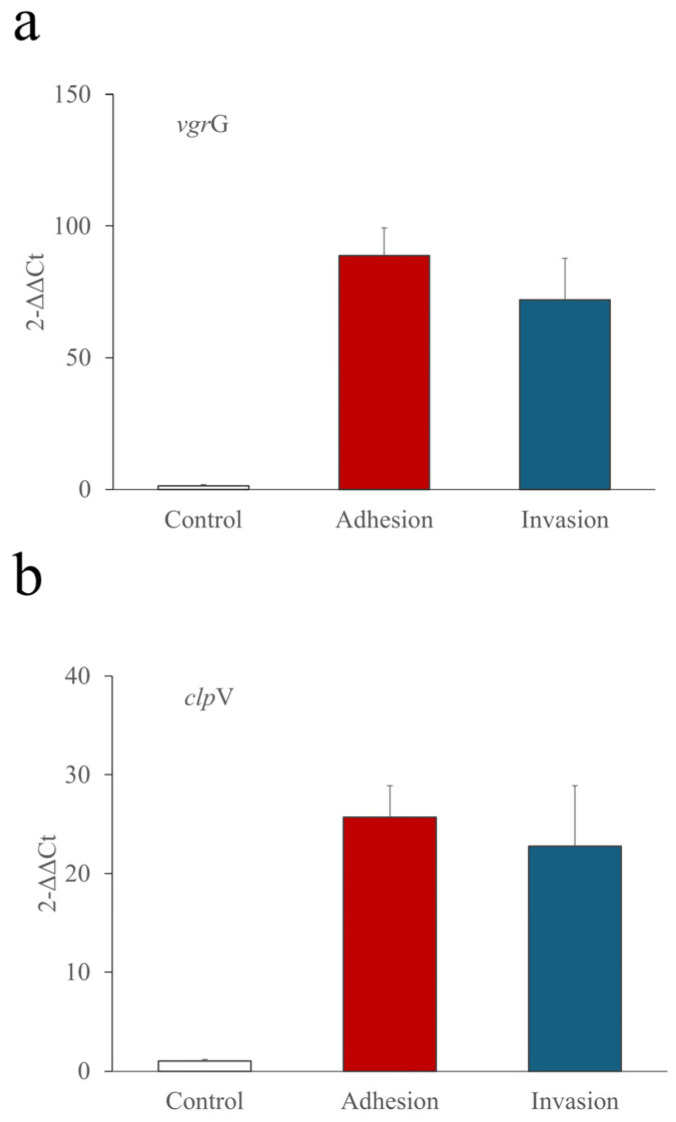
Quantitative PCR confirmed the upregulation of the T6SS genes *vgr*G (**a**) and *clp*V (**b**) upon interaction with the co-culture of Caco-2:HT29-MTX compared to the control. Pure culture of the *E. coli* strain HMLN-1 was used as a control. All tests were carried out in three biological replicates, and the results were expressed as mean ± SEM.

**Table 1 microorganisms-12-01851-t001:** Comparison between the adhesion, invasion and translocation abilities of *E. coli* HMLN-1 and non-translocating strain *E. coli* JM109 (negative control) in a co-culture of Caco-2 and HT29-MTX cell lines (9:1) and their survival and replication in differentiated THP-1 monocytes cells. For each comparison between HMLN and JM109, their *p* values have been given.

Strains	Percentage Colonisation of Co-Culture Cells	No. of Diffusely Adhering Bacteria (CFU) per Cell	No. of Invading Bacteria (CFU) per Well	No. of Intra-Macrophagic Surviving Bacteria (CFU) per Well
HMLN-1	79% ± 1.2%	5.7 ± 0.4	1.1 × 10^4^ ± 1.1 × 10^3^	475 ± 60
*E. coli* JM109	23.3% ± 2.1%	2.0 ± 0.2	1700 ± 245	90 ± 7
*p* value	*p* < 0.0001	*p* < 0.0001	*p* = 0.0012	*p* = 0.0238

**Table 2 microorganisms-12-01851-t002:** PCR detection of virulence genes (VGs) associated with AIEC, iron acquisition and type 6 secretion system (T6SS) among translocating *E. coli* strains HMLN-1 and KIC-1, and non-translocating strains 73–89, 46-4 and JM109.

	AIEC-Associated VGs						Iron Acquisition VGs					T6SS VGs	
*E. coli*	*htr*A	*omp*C	*lpf*A	*dsb*A	*clb*A	*afa*C	*hly*A	*fyu*A	*iut*A	*ire*A	*iro*N	*clp*V	*vgr*G
**HMLN-1**	**+**	**+**	−	**+**	−	**+**	**+**	**+**	−	**+**	**+**	**+**	**+**
**KIC-1**	**+**	**+**	−	**+**	−	**+**	−	**+**	**+**	−	−	−	−
**73–89**	−	**+**	−	−	−	−	−	**+**	**+**	−	−	−	−
**46-4**	−	−	−	−	−	−	−	−	**+**	−	−	−	−
**JM109**	−	−	−	−	−	−	−	−	−	−	−	−	-

**Table 3 microorganisms-12-01851-t003:** Prevalence of type 6 secretion system (T6SS) virulence genes (VGs) among pathogenic *E. coli* strains representing 37 common (C) types and 95 single (S) types isolated from adults with community-acquired UTI (CA-UTI) or diarrhoea, children and adults with hospital-acquired UTI (HA-UTI), adults with inflammatory bowel disease (IBD) and adults with septicaemia.

Source of Infection	*E. coli* Type	No. of C-Types Or s-Types	No. Isolates Represented	*clp*V	*vgr*G	Total (%) Positive
Adult CA-UTI	ExPEC	C1-C11	75	67	67	89%
Child HA-UTI	ExPEC	C12-C25	75	60 **	59 **	83%
Adult septicaemia	ExPEC	C26-C37	74	47	49	66%
Adult HA-UTI	ExPEC	S1-S39	40	17	17	43%
Child diarrhoea	IPEC	S1-S30	30	3 *	3 *	10%
Adult IBD	IPEC	S1-S24	25	8	8	32%

* One S-type positive for both VGs, one S-type positive for only *clp*V and one S-type positive for only *vgr*G. ** Fifty-seven isolates positive for both VGs, three isolates positive for only *clp*V and two isolates positive for only *vgr*G. Extra-intestinal pathogenic *E. coli* strains (ExPEC) vs. intestinal pathogenic *E. coli* (IPEC) strains (*p* < 0.0001); adult septicaemic strains vs. IPEC (*p* = 0.007).

## Data Availability

Data available upon request.

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
