# Peer review of "Identification of Virulence Genes Associated with Pathogenicity of Translocating Escherichia coli with Special Reference to the Type 6 Secretion System"

_microorganisms, 2024, doi:10.3390/microorganisms12091851_

Round 1

Reviewer 1 Report

Comments and Suggestions for Authors

The authors conducted an in-depth analysis of the translocation capacity of E. coli and also compared the occurrence of some virulence genes in an extensive collection of E. coli. Bacteria with this capacity are extremely important to study, and there is a significant lack of information regarding the related mechanisms. Thus, the research presented is highly valuable and was very well executed. The authors analyzed this capacity using two cell lines and also assessed a collection of E. coli in relation to this ability.

The manuscript is very well written, including all sections. However, I suggest that the authors reconsider the presented conclusion, as it may be somewhat confusing for readers who do not read the entire article. I recommend being more precise and objective in the conclusion, avoiding what may seem like a repetition of the main results.

I have two questions regarding the manuscript:

  1. How do the authors know that the E. coli strains evaluated in the study were translocating E. coli and not just ExPEC/UPEC?
  2. Why did the authors not perform a deletion of this gene in the HMLN1 isolate and compare the differences in translocation capacity?

Minor remarks:

  • Line 28 – HMLN1 – is this correct?
  • Line 97 – Standardize the use of °C.
  • Line 100 – Double-check the use of italics for scientific names throughout the manuscript.
  • Lines 109-111 – The description of these isolates could be made available as a supplementary file in the manuscript.
  • In Table 1, italicize the gene names.
  • In Table 2, it would be better to include the p-values in a separate column rather than in the table title.
  • Line 311 – This gene should also be italicized, correct?

Author Response

The authors conducted an in-depth analysis of the translocation capacity of E. coli and also compared the occurrence of some virulence genes in an extensive collection of E. coli. Bacteria with this capacity are extremely important to study, and there is a significant lack of information regarding the related mechanisms. Thus, the research presented is highly valuable and was very well executed. The authors analyzed this capacity using two cell lines and also assessed a collection of E. coli in relation to this ability.

The manuscript is very well written, including all sections. However, I suggest that the authors reconsider the presented conclusion, as it may be somewhat confusing for readers who do not read the entire article. I recommend being more precise and objective in the conclusion, avoiding what may seem like a repetition of the main results.

I have two questions regarding the manuscript:

  1. How do the authors know that the E. coli strains evaluated in the study were translocating E. coli and not just ExPEC/UPEC?

Our response: This is very interesting question. Actually in a concurrent study we are testing all those strains for their possible ability to translocate. Strains included in this study were initially isolated from cases of diarrhea, patients with IBD, or blood stream infection (septicemic patients) and urinary tract infection. Upon isolation of those strains we tested them for the presence virulence genes associated with causing diarrhea, IBD, septicemia and UTI to identify their pathogenic mechanisms. Based on their virulence properties we confirmed those strains to be associated with the disease from which they had been isolated. They were not tested for their ability to translocate. Based on our findings (i.e. presence of T6SS genes), in this study we are now testing them for their ability to translocate. The conclusion of the study (which has also  been revised ) we point at the possible ability of ExPEC strains that carry T6SS-1 to translocate (see the revised conclusion). 

  1. Why did the authors not perform a deletion of this gene in the HMLN1 isolate and compare the differences in translocation capacity?

Our response:

We agree with the reviewer that gene deletion of the various T6SS genes is a valuable experiment to validate our findings, however, performing this experiment was out of the scope of the current project as we focused our attention on the distribution and expression of the T6SS genes. It is certainly within the future planning of this project to perform deletion of various T6SS genes (e.g. clpV or vgrG) to validate our findings that the T6SS confers a translocation advantage in E. coli.  We have included a section in our discussion to refer to this goal as future directions of this project (see lines 444-448 in the revised version).

Minor remarks:

  • Line 28 – HMLN1 – is this correct? 

Our response:

Thank you for picking up on this typo, it has been corrected in-text.

  • Line 97 – Standardize the use of °C.

Our response:

All instances of °C now have a consistent presentation. Those required correction have been highlighted with yellow colour

  • Line 100 – Double-check the use of italics for scientific names throughout the manuscript.

Our response:

We have revised the manuscript and used italics to those instances that were missing italics in our original version.

  • Lines 109-111 – The description of these isolates could be made available as a supplementary file in the manuscript.

Our response: A supplementary table (Supplementary table 1) is already included in the manuscript without a reference to this table. In the revised version we have made this correction and referred to Supplementary table 1 (see supplementary tables and our reference to this table in lines 112-113 in the revised version.

  • In Table 1, italicize the gene names.

Our response: We have italicized genes in table 1 and because this table only contained a list of primers sequence and the PCR protocols, we have now presented that table 2 as Supplementary table 2.

  • In Table 2, it would be better to include the p-values in a separate column rather than in the table title.

Our response: We added p values in a separate row as in some columns there were two comparisons which made it rather confusing if we had added p values in a column. Therefore, significance of the difference between comparison of adhesion and invasion of HMLN-1 and the control strain (i.e. JM109) are now given in a raw below them.

  • Line 311 – This gene should also be italicized, correct?

Response: Thank you for picking up on this, we have corrected the presentation..

Reviewer 2 Report

Comments and Suggestions for Authors

The aim of the study was a very interesting issue of identification of virulence genes associated with pathogenicity of translocating Escherichia coli with special reference to the type 6 secretion system. I suppose a lot of effort went into the study. Novelty, a significant impact of the results as well as several findings made on the basis of the obtained results, are really interesting on a global scale.

My only concerns are as follows:

Although the number of strains included into the study is limited, it would be good to mention  at least that the study will be continued, involving e.g. knock out genes to confirm the observed results.

It would be beneficial if some figures from Results and from the applied methods were included into the manuscript with a better quality (e.g. the legend within the Figure 1) or at all (pictures of your model in a scheme "before and after the research").

Some spelling mistakes and italics are missing within the whole manuscript.

However, the points mentioned above do not decrease the overall value of the research and manuscript.

Author Response

The aim of the study was a very interesting issue of identification of virulence genes associated with pathogenicity of translocating Escherichia coli with special reference to the type 6 secretion system. I suppose a lot of effort went into the study. Novelty, a significant impact of the results as well as several findings made on the basis of the obtained results, are really interesting on a global scale.

My only concerns are as follows:

  • Although the number of strains included into the study is limited, it would be good to mention  at least that the study will be continued, involving e.g. knock out genes to confirm the observed results.

Our response: We agree with the reviewer that gene deletion (gene knockout) of the various T6SS genes is a valuable experiment to validate our findings. This will be our priority and confirms the role of this system in translocation of E. coli. For that reason we have included this aim as future directions of this project in the manuscript (see lines 444-448 in the revised version highlighted in green color)

  • It would be beneficial if some figures from Results and from the applied methods were included into the manuscript with a better quality (e.g. the legend within the Figure 1) or at all (pictures of your model in a scheme "before and after the research").

Our response: We thank the reviewer for detecting the decreased quality of Figure 1 and 2. We have recreated these images to highlight and have provided these with a resolution of 300dpi for clarity.

  • Some spelling mistakes and italics are missing within the whole manuscript.

Our response: We have reviewed the manuscript and made edits where italics were missing and corrected spelling mistakes (highlighted in green).
